# Exploring the Prognostic and Predictive Roles of Ki-67 in Endometrial Cancer

Laura Paleari [1,*], Mariangela Rutigliani [2], Oriana D'Ecclesiis [3], Sara Gandini [3], Irene Maria Briata [4], Tania Buttiron Webber [4], Nicoletta Provinciali [4] and Andrea DeCensi [3,4,5]

1 Research, Innovation and HTA Unit, A.Li.Sa., Liguria Health Authority, 16121 Genoa, Italy
2 Division of Pathology, E.O. Galliera Hospital, 16128 Genoa, Italy; mariangela.rutigliani@galliera.it
3 European Institute of Oncology IRCCS, 20141 Milan, Italy; oriana.decclesiis@ieo.it (O.D.); sara.gandini@ieo.it (S.G.); andrea.decensi@galliera.it (A.D.)
4 Division of Medical Oncology, E.O. Galliera Hospital, 16128 Genoa, Italy; irene.maria.briata@galliera.it (I.M.B.); tania.webber@galliera.it (T.B.W.); nicoletta.provinciali@galliera.it (N.P.)
5 Wolfson Institute of Population Health, Queen Mary University of London, London E1 4NS, UK
* Correspondence: laura.paleari@alisa.liguria.it; Tel.: +39-010-5484243

**Abstract:** Background: Up to now, endometrial cancer (EC) treatments are mainly represented by surgery followed by adjuvant chemotherapy or radiotherapy. The updated guidelines give a 2A recommendation for the use of hormone therapy only in advanced low-grade ECs, underlying the need for more data on the role of hormone therapy in the adjuvant setting. Methods: The clinicopathological data of 158 early-stage EC patients was retrospectively collected. A Ki-67 cut-off value of 40% was established based on literature data. Disease-free survival (DFS) and Overall survival (OS) were evaluated. Results: Results: Multivariate analysis of DFS and OS showed a significantly increased risk of progression in patients with >40% Ki-67 [HR = 3.13 (95% CI; 1.35–7.14); $p = 0.007$] and a significantly higher relative risk of death [HR = 3.70 (95% CI; 1.69–8.33); $p = 0.001$]. The predictive role of the Ki-67 index was highlighted by the clinical benefit of adjuvant hormone in patients with high Ki-67. Conclusions: Our results suggest a positive role of the Ki-67 index as a prognostic and potentially predictive marker in EC, although further studies are warranted to reach a definitive conclusion.

**Keywords:** Ki-67; predictive; prognostic; endometrial cancer; hormone therapy; DFS; OS

## 1. Introduction

In the Western world (Europe, USA), cancer of the endometrium represents the seventh cause of cancer death in women [1]. In the USA, 66,200 new cases are estimated to be diagnosed in 2023, and about 13,000 deaths are expected, with an increase in mortality rates of about 1% per year [1]. In Italy, endometrial cancer is the third most frequent neoplasm in women between the ages of 50 and 69, and the five-year survival rate is around 77% [2,3]. Historically, endometrial cancers (ECs) have been divided into two broad categories based on their anatomo-pathological characteristics: type I (estrogen-dependent) and type II (estrogen-independent) endometrial tumors with different related risk factors [2,4]. With regard to type I EC, there is a common denominator underlying all the risk factors that contribute to the increased probability of developing this type of neoplasm: the presence of a high estrogenic hormonal activity not well balanced by the progestin [5]. It has been shown that the main risk factors can be: (i) environmental, such as overweight, obesity, a high animal fat diet, diabetes, and hypertension; (ii) hormonal, such as early menarche, late menopause, ovarian polycystosis, nulliparity, and hereditary-familial factors. Furthermore, aging has been shown to increase the risk of developing this type of cancer. About 80% of cases of EC occur in postmenopausal women over the age of 60, where most of the

circulating estrogen is derived from the action of aromatase, the enzyme that converts androgens to estrogens [4,6]. Recently, the Cancer Genome Atlas (TCGA) project [7] has led to a new classification of EC according to the molecular and clinicopathological features of the tumor, defining four distinct groups: (i) POLE mut (i.e., polymerase epsilon-ultra mutated), (ii) dMMR/MSI (i.e., mismatch repair deficient/microsatellite-unstable), (iii) NSMP (i.e., no specific molecular profile), and (iv) p53 mut (i.e., frequent pathogenic variants in TP53). These different molecular patterns correspond to different degrees of risk of recurrence [8], providing useful elements to establish the indication and type of a possible adjuvant therapy; therefore, this new classification is now recommended by the most updated guidelines [9,10]. To date, treatment for EC is primarily represented by surgery, followed by adjuvant radiotherapy or chemotherapy [9]. The guidelines for EC management have been updated by the National Comprehensive Cancer Network (NCCN), encompassing hormone therapy for advanced low-grade ECs, with a 2A recommendation due to the lack of conclusive evidence from clinical trials [10]. Recently, we published a retrospective analysis on clinical and pathological factors in stage I–II ECs showing a benefit in terms of PFS and OS in estrogen/progesterone receptor (ER/PgR) positive cancers [11]. The results of our research suggest a potential predictive role for steroid receptors and highlight the need to conduct randomized clinical trials (RCTs) to discover predictive/prognostic markers for this neoplasia [11]. Ki-67 antigen is a nuclear protein closely associated with cell proliferation whose expression is easily visualized in immunohistochemistry (IHC) [12]. It can be found exclusively within the nucleus during the interphase, and since it is present during all phases of the cell cycle (G1, S, G2, and mitosis) but is absent in the G0 phase, Ki-67 represents a useful marker of the so-called growth fraction of a given cell population [12]. In the clinical setting, the Ki-67 index—which represents the portion of antigen-positive tumor cells—is often correlated with a poor prognosis of the clinical course of neoplastic disease [13,14]. Breast cancer (BC) is the first cancer in which, since the early 1980s, a potential role of Ki-67 has been shown in the prognosis of the disease [15]. To date, the debate on its validity as a prognostic marker in BC is still open, while its value is recognized only for prognosis assessment in ER-positive and HER2-negative patients to identify those who do not need adjuvant chemotherapy [16,17]. Last year, at the American Society of Clinical Oncology (ASCO) congress, the results of the LUMINA study were presented. The study assessed the efficacy of radiotherapy in women with early-stage type-A luminal BC, which encompasses approximately half of all diagnostic BCs in this category. From the analysis, it emerged that in women expressing low levels of Ki-67 on cancer cells, radiotherapy can be avoided without changing the probability of relapse, thus preserving women from any side effects [18]. Over time, several studies have tried to evaluate the role of Ki-67 as a prognostic factor in EC with conflicting results, and some of these have validated it as a predictor of response to treatment [19–22]. In fact, unlike in BC, where Ki-67 has been internationally recognized as a prognostic biomarker with a threshold of >20% to drive clinical choice [23], for EC there are no guidelines standardizing Ki-67 measurement and its clinical relevance as a biomarker of response. Interestingly, Kitson and colleagues, using the Ki-67 in BC Working Group guidelines, evaluated 179 EC biopsies, finding a positive correlation between grade, stage, depth of myometrial invasion, and the score of the proliferation index ($p < 0.03$) [24]. In their conclusions, the authors declared that further studies are warranted to determine Ki-67 as a biomarker of treatment response [24]. Recently, a study was published in stage I-II EC patients to determine a cut-off value for Ki-67 detected with IHC for predicting disease recurrence [25]. The authors found an ideal cut-off of 38% for predicting the recurrence in stage I–II ECs, suggesting that Ki-67 might become a contributor to customized adjuvant treatment, particularly for early FIGO stages and low-risk ECs [25]. To date, it is a high research priority to establish an international standardized procedure for using Ki-67 in clinical practice to improve the efficacy of EC treatment and identify high-risk patients. Thus, we performed a retrospective analysis on a cohort of EC patients to explore the prognostic and predictive role of Ki-67 and contribute to translating our findings into improved patient outcomes.

## 2. Results

### 2.1. Study Population

In total, 158 patients met our inclusion criteria for potentially curable stage (I–III) EC. The overall median age was 75 years (IQR, 69–83). The median expression levels of steroid receptors were 80% both for ER (IQR, 60–90) and PgR (IQR, 40–90) and 40% (IQR, 30–70) for Ki-67. Overall, 57.6% were at low risk and 42.4% were at high risk, according to the ESMO guidelines [26]. Furthermore, 38.3% of the cohort population had a healthy weight, 28.3% were overweight, and 33.3% were obese. Adjuvant treatment included aromatase inhibitors (AIs:exemestane or letrozole, 71% and 29%, respectively). The choice of the AIs was based on patient preference after medical counseling, based on the lack of sound efficacy data and the known toxicity profile of each compound, and was offered to all the patients [27,28]. The main patient and tumor characteristics are summarized in Table 1. There was no significant difference in terms of patient characteristics among the three treatment groups.

**Table 1.** Patients' characteristics.

| | | Overall (*n* = 158) |
|---|---|---|
| Age, median (IQR) | | 75 (69–83) |
| BMI, median (IQR) | | 26.7 (23.6–31.0) |
| Grade, *n* (%) | G1 | 9 (5.7) |
| | G2 | 82 (52.2) |
| | G3 | 65 (41.4) |
| Stage, *n* (%) | I | 127 (81.4) |
| | II | 13 (8.4) |
| | III | 14 (9.0) |
| Therapy, *n* (%) | Aromatase inhibitors | 92 (59.0) |
| | No adjuvant therapy | 49 (31.4) |
| | Chemotherapy /Radiotherapy | 15 (9.6) |
| ER, median (IQR) | | 80 (60–90) |
| PgR, median (IQR) | | 80 (40–90) |
| Ki-67, median (IQR) | | 40 (30–70) |

IQR, interquartile range; BMI, body mass index; ER, estrogen receptor; PgR, progesterone receptor.

### 2.2. Univariate and Multivariate Analysis of Disease-Free Survival and Overall Survival

At the time of the present analysis, 122 out of 158 (77.2%) patients were alive. The univariate curves showed an augmented risk of death ($p = 0.007$) and of disease progression ($p = 0.05$) in the highly expressing Ki-67 cancers relative to the low Ki-67 group (Figure 1). The Cox regression model adjusted for age and stage identified a significant interaction in the two scenarios, DFS and OS (Table 2). There was a significantly lower risk of progression in the hormone-therapy (HT)-treated high-Ki-67 patients compared with the non-HT-treated group [HR = 0.28 (95% CI; 0.12–0.69), $p = 0.006$]. This relationship did not reach statistical significance for OS ($p = 0.07$). Among patients not treated with HT, those with high levels of Ki-67 (>40%) compared with those with low levels of Ki-67 (<40%) had a more than threefold increased risk of progression [HR = 3.13 (95% CI; 1.35–7.14); $p = 0.007$] and death [HR = 3.70 (95% CI; 1.69–78.33); $p = 0.001$]. A trend toward lower risk of progression [HR = 0.45 (95% CI; 0.19–1.06); $p = 0.068$] and death [HR = 0.26 (95% CI; 0.09–0.76); $p = 0.01$] is identified in the HT-treated patients with low Ki-67 expression compared with patients with no HT and high Ki-67 expression.

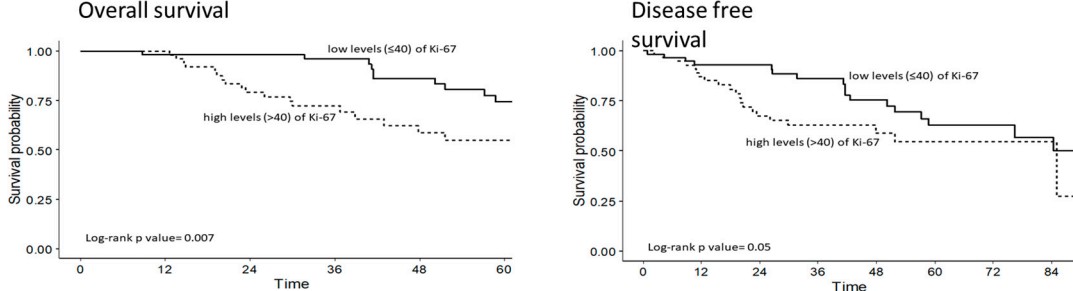

**Figure 1.** Overall survival in months (**left**) and Disease-free survival in months (**right**) in EC patients by Ki-67 expression level; HR adjusted for age, BMI, grading, ER and PgR expression levels, and type of therapy.

**Table 2.** Cox regression model for interaction for Disease-Free Survival and Overall Survival.

| | DFS<br>HR (95% CI), *p* Value | OS<br>HR (95% CI), *p* Value |
|---|---|---|
| Age | 1.05 (1–01–1.08), 0.005 | 1.04 (1.00–1.09), 0.05 |
| Stage III/IV vs. Stage I/II | 7.60 (3.14–18.4), <0.001 | 1.85 (0.56–6.10), 0.31 |
| HT in high Ki-67 vs. no HT in high Ki-67 | 0.28 (0.12–0.69), 0.006 | 0.39 (0.14–1.08), 0.07 |
| No HT in high Ki-67 vs. no HT in low Ki-67 | 3.13 (1.35–7.14), 0.007 | 3.70 (1.69–8.33), 0.001 |
| HT in low Ki-67 vs. no HT in high Ki-67 | 0.45 (0.19–1.06), 0.068 | 0.26 (0.09–0.76), 0.01 |

HT, hormone therapy.

Given the indication for an interaction, as exploratory analyses, we stratified the analyses by Ki67, and we found a statistically significant benefit in terms of DFS [HR = 0.25 (95% CI; 0.09–0.69), *p* = 0.008] and OS [HR = 0.30 (95% CI; 0.10–0.87), *p* = 0.03] in the high Ki-67 group treated with HT and the opposite in the low Ki-67 group even if the association was not significant [HR = 1.65 (95% CI; 0.62–4.42, *p* = 0.32) and HR = 1.07 (95% CI; 0.39–2.91, *p* = 0.89), for DFS and OS, respectively] (Figure 2).

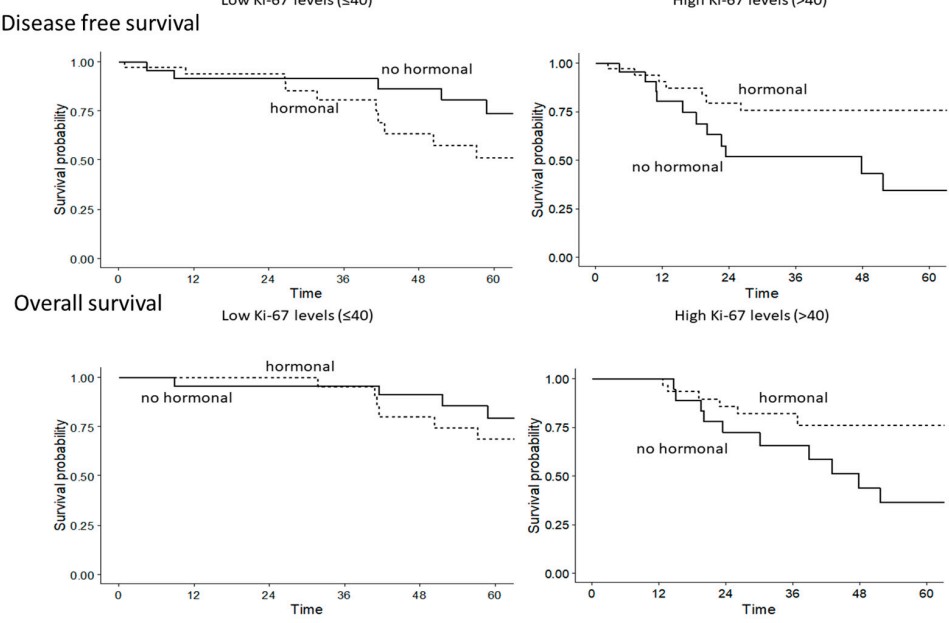

**Figure 2.** Disease-free survival by type of therapy in EC patients with low- and high Ki-67 expression levels; and overall survival by type of therapy in EC patients with low- and high Ki-67 expression levels; HR adjusted for age, stage, and type of therapy.

## 3. Discussion

EC represents almost all cancers affecting the body of the uterus and ranks fifth among the most frequently diagnosed cancers in women (5 percent of all cancer diagnoses in women) [2,3]. These tumors mainly affect adult women after menopause, with a peak incidence after 50 years of age. The increase in diagnoses is linked to a lengthening of the average life span and a change in habits and behaviors (particularly in the diet, which is often too rich in animal fats). Several risk factors have been identified: the presence of an estrogenic activity not adequately balanced by progesterone, age, obesity, diabetes mellitus, and hypertension, which increase the risk of developing cancer by about 3–4 times compared to the general population [2,3]. Nowadays, hysterectomy is the main treatment for EC, followed by chemotherapy and/or radiotherapy for potentially curable EC. One of the priorities in the management of EC is to find biological markers to refine the diagnosis and better drive the therapeutic choice. As mentioned, the four molecular classes of EC also correspond to different classes of risk of recurrence. According to this new classification, the best strategy for determining the most suitable type of adjuvant therapy is given by an algorithm that starts with the determination of the POLE status, then the microsatellite status, and finally p53 [8]. The Ki-67 proliferation index is known to be correlated with a poor prognosis in neoplastic disease [13,14]. It has been demonstrated that radiotherapy can be avoided in patients expressing low Ki-67 levels without perturbing their clinical outcome [18]. Moreover, its value is recognized in ER-positive and HER2-negative patients to identify those who do not need adjuvant chemotherapy [16,17]. Recently, several studies highlighted indications of the potential prognostic/predictive role of Ki-67 in EC management but without a definitive conclusion [13–17,19,20,22,24,25]. A recent study also tried to identify a correlation between risk groups according to the new classification of EC and several biomarkers, including Ki-67, without, however, finding statistically significant variability [29]. Here, we present the results of a retrospective analysis to explore the potential role of Ki-67 as a predictive marker of HT use. In total, we analyzed 158 EC patients with highly expressing ER/PgR cancers previously treated with chemotherapy, radiotherapy, or HT as adjuvant treatment. Patients with highly expressed Ki-67 ECs had an augmented risk of death ($p = 0.007$) and disease progression ($p = 0.05$). Multivariate analysis of DFS and OS showed a trend toward an increased risk of progression in patients with >40% Ki-67 [HR = 1.43 (95% CI; 0.72–2.83); $p = 0.31$] and a more than doubled relative risk of death [HR = 2.50 (95% CI; 1.26–4.95); $p = 0.008$]. The Ki-67 category was also predictive of HT benefit, with a significantly greater magnitude of effect in women with high Ki-67 versus low Ki-67 groups. The major limitation of our study is represented by the small sample size; hence, the results regarding the interaction should be treated with caution. Although the stated statistical power of our study is limited, the results suggest a positive role for the Ki-67 index as a prognostic and predictive marker of response to HT in early-stage EC. Further investigations are warranted to confirm the trends highlighted in this research.

## 4. Materials and Methods

### 4.1. Patients

This is a retrospective cohort study conducted on 158 women operated on for EC at Galliera Hospital. The study was approved by the local ethical committee (code 214-2018, 25 March 2019). Demographic, clinic–pathologic, and follow-up data were obtained from patients' medical records. The decision to undergo an aromatase inhibitor (AI) or no treatment in women with stage I EC was based on patient preference after careful medical counseling by a single medical oncologist (ADC). The inclusion criteria were: (1) histologically diagnosed EC and positive hormone receptor expression; (2) patients operated for EC at Galliera Hospital; (3) well-differentiated tumor (International Federation of Obstetrics and Gynecology classification) [26]; (4) age > 18 years; (5) no contraindication to AIs use, including any prior cancer, prior cardiovascular disease, osteoporosis, grade 2 or higher biochemical alterations, prior use of selective estrogen receptor modulators or AIs, or mental disorders [16].

### *4.2. Tissue Sampling, Histopathological Analysis, and Ki-67 Determination*

Tumor sampling was performed immediately after the hysterectomy. For histopathologic examination, 2-micrometer-thick FFPE sections were stained with the conventional hematoxylin and eosin stain. Two pathologists confirmed the diagnosis of endometrial carcinoma. IHC was performed on 2 μm sections with an automated IHC staining system (Ventana BenchMark ULTRA, Ventana Medical Systems, 20900 Monza, Italy). All cases were stained with IHC Ki-67/MIB1 (clone 30-9, 1/100, Ventana). Glands and endometrial stroma constituted the internal positive control for the IHC procedure. Two independent observers scored the slides. Figure 3 shows the IHC evaluation of Ki-67 in high (≥40%) and low (<40%) expression EC samples.

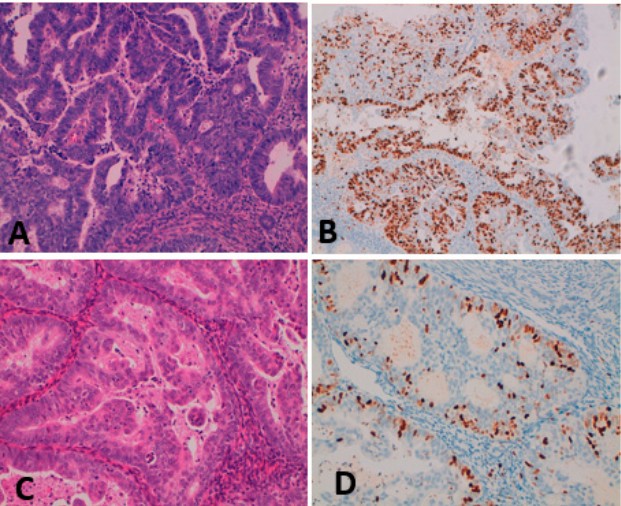

**Figure 3.** (**A**,**B**) High-grade EC Hematoxylin-eosin staining magnification 20×; the Ki-67 proliferation index is 70% (Ki-67 Mib1, magnification 10×); (**C**,**D**) Low-grade EC Hematoxylin-eosin staining magnification 20×; the Ki-67 proliferation index is 15% (Ki-67 Mib1, magnification 20×).

### *4.3. Evaluation of Ki-67 Value*

DFS is defined as the length of time after primary treatment for a cancer ends that the patient survives without any signs or symptoms of that cancer. OS is defined as the period from the start of treatment administration to death, or the last follow-up.

### *4.4. Statistical Analysis*

Descriptive statistics were used to present patients and tumor characteristics. Data were presented as relative frequencies (percentage) or medians and interquartile ranges (IQRs) for continuous variables. Ki-67 was analyzed as a categorical variable, considering the median. Disease-free survival (DFS) was calculated from first treatment to disease progression or death (event) or last follow-up (censored). Overall survival (OS) was calculated from the first treatment to death (event) or the last follow-up (censored). DFS and OS curves were estimated with the Kaplan-Meier method, and survival distributions were compared using the log-rank test. Multivariate Cox proportional hazard models were used to investigate the independent prognostic role of Ki-67, adjusting for other significant prognostic factors and confounders. Subgroup analyses by Ki-67 levels were conducted. Time-dependent covariates were used in the models to account for the fact that not all subjects had started therapy since the time of observation. Results are presented as hazard ratios (HRs) with 95% confidence intervals (95% CIs). For all analyses, a two-tailed $p < 0.05$ was considered statistically significant. The statistical analyses were performed with R software, version 4.2.1.

**Author Contributions:** L.P.: conceptualization, methodology, investigation, writing—original draft, writing—review and editing, project administration, and supervision. M.R.: conceptualization, writing—review and editing. O.D.: statistical analyses, review, and editing S.G.: statistical analyses, review, and editing. I.M.B.: data entry, review, and editing. T.B.W.: data entry, review, and editing. N.P.: writing—review and editing. A.D.: supervision, project administration, and funding acquisition. All authors have read and agreed to the published version of the manuscript.

**Funding:** This work was supported by the Italian Association for Cancer Research (AIRC), grant IG 2018, ID 21534.

**Institutional Review Board Statement:** The study was conducted in accordance with the Declaration of Helsinki, and approved by the Liguria Ethical Committee code 214-2018, 25 March 2019.

**Informed Consent Statement:** Informed consent was obtained from all subjects involved in the study.

**Data Availability Statement:** The data presented in this article is available on reasonable request from the corresponding author.

**Acknowledgments:** To Rami and Violetto, without whom nothing would be possible.

**Conflicts of Interest:** The authors declare no conflict of interest.

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
