# Peer review of "Exploring the Prognostic and Predictive Roles of Ki-67 in Endometrial Cancer"

_2673-8937, doi:10.3390/ijtm3040033_

Round 1

Reviewer 1 Report

Comments and Suggestions for Authors

This is a single-center retrospective study evaluating the possible prognostic and predictive role of Ki-67 in endometrial cancer (EC).

The paper is well written, and the English language is appropriate and understandable.

The clinical arguments presented are interesting. The recently updated guidelines by International Cancer Societies encompass hormone therapy for advanced low-grade ECs. Furthermore, to date, few data are available for using Ki-67 in clinical practice in order to improve the efficacy of EC treatment.

The results of this study support the predictive role of Ki-67 on the outcome of women with EC identifying high-risk patients with the potential benefit of hormonal therapy.

In any case, the limitations and bias of this study are correctly reported by the Authors.

Specific comments:

Could the Authors report further pathological data? All the patients enrolled had type I EC?  

The Authors reported that inclusion criteria were potentially curable stages (I-III) (Row No. 103). Table 1 shows two patients at stage IV.

How many patients showed Ki-67 value of more than 40%?

The Authors reported that 57.6% and 42.4% were low and high-risk patients, respectively (Row No. 106), according to the new ESMO classification. Could the Authors give data on molecular characteristics (MMR/MSI, No Specific Molecular Profile, p53 mut, and POLE mut)?

The Author reported (Rows No. 169) that a recent study failed to identify a significant correlation between different risk groups according to the new classification of EC and several biomarkers including  Ki-67. Did the multivariate analysis take the different risk groups into consideration?

Author Response

Dear,

below our point by point reply to your concerns:

1- Could the Authors report further pathological data? All the patients enrolled had type I EC? 

We have take into account in the Table the main caractheristics usefull for our scope. We are ongoing to perform other evaluations on, for instance, the molecular profile to confirm and refine our analyses. The population in our analyses encompassed type I-III EC (Table 1)

2- The Authors reported that inclusion criteria were potentially curable stages (I-III) (Row No. 103). Table 1 shows two patients at stage IV. You are right we have changed the table.

3- How many patients showed Ki-67 value of more than 40%? The 64% of patients have a ki-67 > 40%

4- The Authors reported that 57.6% and 42.4% were low and high-risk patients, respectively (Row No. 106), according to the new ESMO classification. Could the Authors give data on molecular characteristics (MMR/MSI, No Specific Molecular Profile, p53 mut, and POLE mut)?

As previously stated we are performing these evaluations to validate our explorative assessment.

5- The Author reported (Rows No. 169) that a recent study failed to identify a significant correlation between different risk groups according to the new classification of EC and several biomarkers including  Ki-67. Did the multivariate analysis take the different risk groups into consideration? Yes we have taken into account these in the multivariate analyses.

Reviewer 2 Report

Comments and Suggestions for Authors

The paper “Exploring the Prognostic and Predictive Role of Ki67 in Endometrial Cancer” analyses Ki67 expression in 158 EC patients and assess its ability to be used as a prognostic and predictive marker for diseases free survival and overall survival in the context of adjuvant hormone therapy. Besides some grammatical errors (English language needs a little bit of work), the paper is well written. The introduction clearly presents the background literature to the problem, while the discussion explains how their work contributes to the field. I really enjoyed reading this paper and thank the authors for an interesting study. A few specific comments are as follows:

1. Line 108: “Adjuvant treatment included AIs”. There is no definition for an AI in the manuscript.

2. Line 122: “HT-treated”. Again, there is no definition for HT in the text. Please ensure all abbreviations are described properly first before their continual use throughout the body of the text.

3. Table 2: The rows of the table do not align properly, making it difficult to know which HR values correspond to which comparison.

4. It would be nice to show an example of histology images for a low Ki67 and a high Ki67 expression tissue for EC.

Comments on the Quality of English Language

There are some changes to improve the quality of English language in this paper. For example, "To date the debate on its validity as prognostic markers in BC us still open" could be written as "To date, there is ongoing debate as to the validity of Ki67 as a biomarker in breast cancer". 

Full stops should also be after the reference at the end of the sentence, not before.

Author Response

  1. Line 108: “Adjuvant treatment included AIs”. There is no definition for an AI in the manuscript. Done
  2. Line 122: “HT-treated”. Again, there is no definition for HT in the text. Please ensure all abbreviations are described properly first before their continual use throughout the body of the text. Done
  3. Table 2: The rows of the table do not align properly, making it difficult to know which HR values correspond to which comparison. we have change the table according your suggestion.
  4. It would be nice to show an example of histology images for a low Ki67 and a high Ki67 expression tissue for EC. We have added the Figure 3